

# Organic Carbon, Mercury, and Sediment Characteristics along a land
# – shore transect in Arctic Alaska
Frieda P. Giest[1,2], Maren Jenrich[1,3*], Guido Grosse[1,3], Benjamin M. Jones[4], Kai Mangelsdorf[5], Torben
Windirsch[1,a], Jens Strauss[1*]
[1]Alfred Wegener Institute, Helmholtz Centre for Polar and Marine Research, Permafrost Research Section, Potsdam, Germany
[2]University of Potsdam, Institute of Environmental Science and Geography, Potsdam, Germany
[3]University of Potsdam, Institute of Geosciences, Potsdam, Germany
[4]Institute of Northern Engineering, University of Alaska Fairbanks, Fairbanks, Alaska, USA
[5]German Research Centre for Geosciences GFZ, Helmholtz Centre Potsdam, Organic Geochemistry Section, Potsdam,
Germany
[a] now at Research Institute for Sustainability Helmholtz Centre Potsdam, Potsdam, Germany
*Correspondence to*: Maren Jenrich (maren.jenrich@awi.de) and Jens Strauss (jens.strauss@awi.de)
**Abstract.** Climate warming in the Arctic results in thawing permafrost and associated processes like thermokarst, especially
in ice-rich permafrost regions. Since permafrost soils are one of the largest organic carbon reservoirs of the world, their thawing
could lead to the release of greenhouse gases, further exacerbating climate warming. To enhance predictions of potential future
impacts of permafrost thaw, we studied how soil characteristics change in response to permafrost landscapes affected by
thermokarst processes in an Arctic coastal lowland. We analysed six sediment cores from the Arctic Coastal Plain of northern
Alaska, each representing a different landscape feature along a gradient from upland to thermokarst lake and drained basin to
thermokarst lagoons in various development stages. For the analysis, a multiproxy approach was used including
sedimentological (grain size, bulk density, ice content), biogeochemical (total organic carbon (TOC), TOC density (TOCvol),
total nitrogen (TN), stable carbon isotopes ($\delta^{13}$C), TOC/TN ratio, mercury (Hg)), and lipid biomarker (*n*-alkanes, *n*-alkanols,
average chain length (ACL), $P_{aq}$, $P_{wax}$, carbon preference index (CPI), higher plant alcohol index (HPA)) parameters. The
results showed highest TOC contents in samples of the thermokarst lake and the drained thermokarst lake basin. Lowest TOC
contents were measured in the samples of the semi-drained thermokarst lagoon. The comparison of unfrozen and frozen
deposits showed significantly higher TOCvol and TN in the unfrozen deposits. Indicated by the ACL, $\delta^{13}$C and the $P_{aq}$, $P_{wax}$
we found a stronger influence of aquatic organic matter (OM) in the OM composition in the soils covered by water, compared
to those not covered by water. Moreover, it was indicated by the results of the $\delta^{13}$C, TOC/TN ratio, and the CPI that the saline
deposits contain stronger degraded OM than the deposits not influenced by saltwater. Additionally, we found positive
correlations between the TOC and TOCvol and the Hg content in the deposits. The results indicate that thermokarst-influenced
deposits tend to accumulate Hg during thawed periods and thus contain more Hg than the upland permafrost deposits that have
not been impacted by lake formation. Our findings offer valuable insights into the dynamics of carbon storage and vulnerability
to decomposition in coastal permafrost landscapes, reflecting the interplay of environmental factors, landform characteristics,
and climate change impacts on Arctic permafrost environments.
## 1 Introduction
Climate warming represents one of the most pressing global environmental challenges of our time. Arctic regions are currently
changing rapidly, since they experience some of the highest rates of impacts from climate change (Intergovernmental Panel
on Climate Change (IPCC), 2022, 2023). Surface air temperatures in the Arctic increased up to four times the rate the global
mean air temperature did over the last decades, a phenomenon referred to as Arctic amplification (Ballinger et al., 2023; Cohen



et al., 2020; Rantanen et al., 2022). The local drivers of this amplification include the decrease of sea ice and snow cover,
resulting in a decreased albedo, and a shift of cloudiness over the Arctic (Ballinger et al., 2023). Moreover, there are remote
drivers which contribute to the amplification, including an increased total water vapour in the Arctic atmosphere, due to an
increased evapotranspiration and atmospheric moisture transport from the mid-latitudes and tropics, and accelerated heat from
the atmosphere and the ocean (Cohen et al., 2020). As a result, surface temperatures in the Arctic during the winters in 2016
and 2018 were 6 °C above the average temperatures between 1981–2010 (Intergovernmental Panel on Climate Change (IPCC),

46  2022).

One impact of this warming is the thaw of permafrost, which underlies large areas of the Arctic (Biskaborn et al., 2019; Smith
et al., 2022). In some locations a total increase of 2–3 °C in the last 30 years was found within 10–20 m soil depth. Permafrost
has been identified as a large and vulnerable reservoir of organic carbon (OC) and due to climate change is considered a
potential major future carbon source in the earth system (Hugelius et al., 2014; Mishra et al., 2021; Schuur et al., 2022). It is
estimated that terrestrial deposits in permafrost regions store approximately 1460–1600 Gt carbon, which is about twice as
much as is currently present in the atmosphere (Schuur and Mack, 2018; Strauss et al., 2024a). As permafrost thaws, the soils
can turn from a carbon sink to a carbon source (Schuur et al., 2009). Increased temperatures cause an acceleration of microbial
activity and thus an increased decomposition of organic carbon in the deposits, leading to the release of greenhouse gases in
the form of carbon dioxide and methane with the potential to further exacerbate climate change (Miner et al., 2022). In order
to analyse the quality of organic matter (OM) in the different soils lipid biomarkers can be used. Indices like the average chain
length of $n$-alkanes (ACL), the carbon preferences index (CPI), and the higher plant index (HPA) can provide information
about the source of the OM, as well as the level of degradation (Jongejans et al., 2020, 2021; Strauss et al., 2015).
Another consequence of permafrost thaw is the change of the landscape, for example due to melting ground ice causing surface
subsidence and the development of thermokarst features (Grosse et al., 2013; Kokelj and Jorgenson, 2013). Around 20 % of
the permafrost regions are affected by thermokarst processes, including the formation of thermokarst lakes and drained lake
basins (Grosse et al., 2013; Jones et al., 2022; Olefeldt et al., 2016). In a coastal environment, increased coastal and riverbank
erosion, sea level rise, higher water temperatures, and a reduced sea ice cover can lead to the inundation of thermokarst lakes
and drained thermokarst lake basins by ocean water and the formation of thermokarst lagoons (Jenrich et al., 2021;
Schirrmeister et al., 2018). These features add another complex setting of biogeochemical and hydrochemical processes in the
transitional stage between terrestrial and marine environments, to the already diverse thermokarst landscapes (Schirrmeister et
al., 2018).
In addition to the influence of permafrost thaw and the formation of thermokarst features on organic carbon characteristic in
the permafrost and thawed soils, changes in other biogeochemical characteristics may also occur, e.g. through the relocation
and release of mercury (Hg). It was found that considerable amounts of Hg accumulated in the ice-rich permafrost region
(Rutkowski et al., 2021). Since permafrost soils sequestered Hg bound in organic matter over centuries, it is estimated that the
amount of Hg retained in permafrost regions is twice as high as in all other soils, the atmosphere, and the ocean combined
(Schuster et al., 2018). Therefore, Hg is a notable environmental concern in the Arctic region for both humans and wildlife, as
elevated exposure can impact human health and have negative effects on the ecosystems (Rydberg et al., 2010; Smith-Downey
et al., 2010).
In this study, we use a multiproxy approach to characterise OC in different landscape features of a coastal permafrost lowland
along a gradient from upland to thermokarst-affected terrains (lakes and drained basins) to thermokarst lagoons representing a
transition from terrestrial to marine environments on the Arctic Coastal Plain of northern Alaska. We aim to answer how OC
characteristics and correlating biogeochemical parameters change with permafrost degradation and coastal saltwater
inundation.



**2 Study area and study sites**
The study area is located in the Arctic coastal plain of northern Alaska, north of the Teshekpuk Lake (figure 1). The North
Slope, an area framed by the Brooks Range in the south and the Beaufort Sea in the north, encompasses a diverse geology
including deposits originated in the North American craton, passive margin sediments, rift sediments, pelagic sediments,
volcaniclastics and deposits from the foreland basin (Jorgenson et al., 2011). Surface deposits in the study area consist of
glacio-marine silts, marine sands, alluvial sands and silts from the Holocene and mid-Quaternary epochs (Jorgenson and
Grunblatt, 2013).

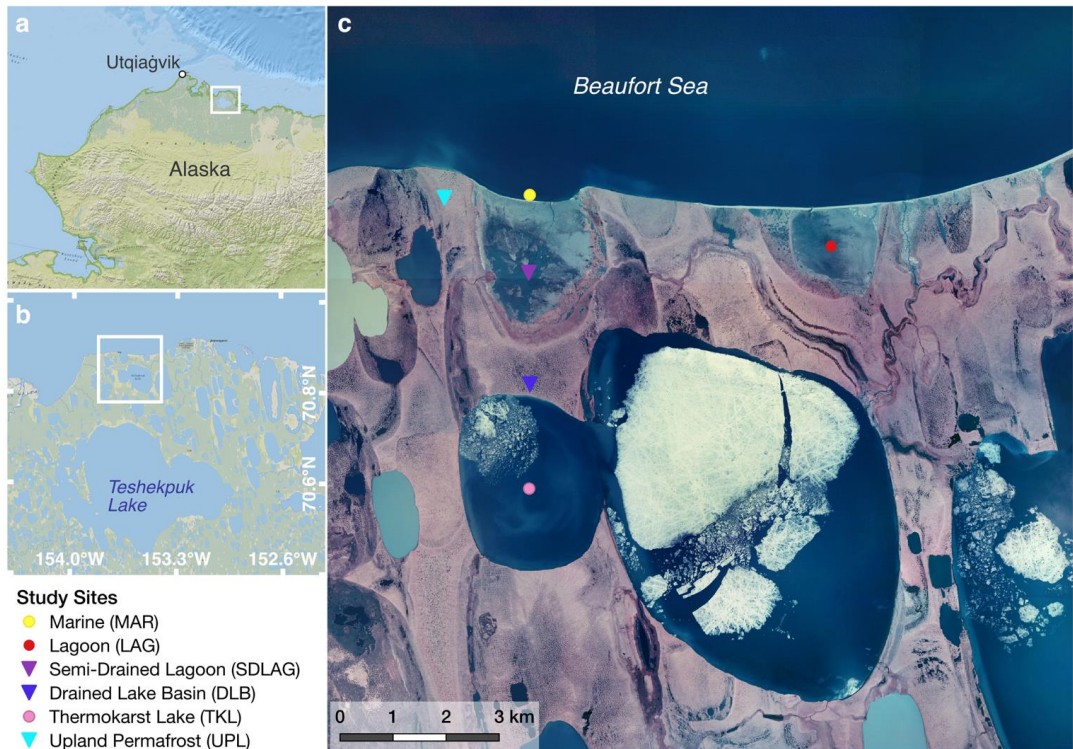


**Figure 1: Map of the study sites located north of the Teshekpuk Lake in northern Alaska.** (a, b) Close-up of the study area, with the
coring locations marked as dots (unfrozen deposits) and triangles (frozen deposits) (c). Sources: a, b: ESRI, c: Color infrared ortho aerial
image (U.S. Geological Survey, Earth Explorer, 2002).
The climate in the region is cold and arid, with a mean annual temperature of -12 °C and a mean annual precipitation of 115 mm
per year (Jorgenson et al., 2011). The soil composition in the area is intrinsically tied to the presence of continuous permafrost,
with an interplay of low temperatures, impeded drainage, freeze-thaw dynamics, cryoturbation, and ground ice aggregation,
collectively shaping its characteristic. The presence of 200 to 400 m thick continuous permafrost also led to the formation and
preservation of one of the largest wetland complexes in the Arctic, which despite the cold and arid climate also lead to the
accumulation of high OC contents in the soils (Jorgenson et al., 2011; Wendler et al., 2014). Moreover, the landscape is
continuously transformed by thawing permafrost and melting ground ice, leading to ground subsidence and the formation of
numerous thermokarst lakes and drained lake basins (Arp et al., 2011; Fuchs et al., 2019; Jones and Arp, 2015; Jorgenson and
Shur, 2007; Wolter et al., 2024). Coastal erosion along the Beaufort Sea coast in this area is among the highest observed in the
Arctic, resulting in the drainage of lakes and formation of thermokarst lagoons and embayments, and is currently accelerating
further (Jones et al., 2009, 2018; Jones and Arp, 2015).



### 3 Material and Methods

#### 3.1 Fieldwork

The fieldwork was performed during a joint German-US expedition to the Teshekpuk Lake area in Alaska in April 2022. For this study six soil cores were selected following a transect from inland to coast, with all core sites being located in close distance to each other (Figure 1). All sample sites represent different landscape features of a coastal thermokarst affected permafrost landscape scape, with the chosen transect describing the transformation pathway from a terrestrial permafrost landscape into a marine environment, following thaw and erosion processes (Jenrich et al., 2021). Three of the cores were frozen: from a permafrost upland (UPL; length 203 cm), a drained thermokarst lake basin (DLB; length 219 cm), and a semi-drained lagoon (SDLAG; length 183 cm). Three other cores were unfrozen: from a thermokarst lake (TKL; length 50 cm), a thermokarst lagoon (LAG; length 31 cm), and marine deposits (MAR; length 12 cm) (figure S1 in the supplements). For reference, all subsample depths are given in centimetres below surface level (cm b.s.l.). The unfrozen sediment cores were sampled using a Push Corer [Ø 6 cm], the frozen sediment cores were sampled using a SIPRE Corer [Ø 7.6 cm]. The frozen sediment cores were kept frozen, while thawed samples were packed and cooled for transport to AWI Potsdam for further analysis.

#### 3.2 Laboratory analysis

In the laboratory, a multi-proxy approach was applied, including sedimentological, biogeochemical and lipid biomarker analysis. The cores were subsampled in intervals of 5 to10 cm. For the biomarker analysis, three to four samples of the longer, frozen cores and one to two samples of the shorter, unfrozen cores were selected, evenly distributed over the length of the cores. In preparation for further analysis all samples were freeze-dried and weighed before and after this process. A more detailed description of the methods used in the laboratory is given in the supplements (Sect. S1 in the supplements).

##### 3.2.1 Sedimentological analysis

The sedimentological analysis included the measurement of water-/ice content, bulk density, and grain size composition.

The water-/ice content was calculated as the difference between wet and dry weight of each sample.

The bulk density (BD) was calculated using equation 1 (Strauss et al., 2012), where the porosity ($n$) of the soil was calculated as the ratio of the pore volume and the total volume of the samples. It was assumed that samples with a water-/ice content of $\geq 20$ % were water-/ice saturated (Strauss et al., 2012), thus the water-/ice content equals the pore volume. Moreover, an ice density at -10 °C of 0.918 g cm$^{-3}$ and a water density at 0 °C of 0.999 g cm$^{-3}$ was used (Harvey, 2019). The dry mineral density ($\rho_s$) was considered to be 2.65 g cm$^{-3}$ (Rowell, 1994).

$$BD = (n - 1) \cdot (-\rho_s) \tag{1}$$

Grain size distribution (GSD) measurement was carried out using a Malvern Mastersizer 3000 with a Malvern Hydro LV wet-sample dispersion unit, measuring in a range between 0.01–1000 µm. All grain size statistics were calculated using the software GRADISTAT (Blott and Pye, 2001).

##### 3.2.2 Biogeochemical analysis

For biogeochemical analysis, all samples were homogenised using a planetary mill [FRITSCH pulverisette 5]. The determination of the total organic carbon content (TOC) was carried out using an ELEMENTAR soliTOC cube elemental analyser, measuring TOC and total inorganic carbon (TIC) via pyrolysis and gas analysis. Using a temperature ramping program to distinguish between TOC and TIC, the device was heated to 400 °C for 230 seconds (TOC), and subsequently



heated to 600 °C for 120 seconds (TIC). Third heating stage was 900 °C for 150 seconds to ensure complete combustion of
inorganic carbon compounds.
The carbon density (TOCvol) of each sample was determined using the bulk density and the TOC content. It was calculated
using the following equation (2) (Strauss et al., 2015).
$$TOC_{vol}[kg\ m^{-3}] = BD[kg\ m^{-3}] \cdot \frac{TOC\ [wt\%]}{100} \qquad (2)$$
The total nitrogen (TN) content was measured using an ELEMENTAR rapid MAX N exceed elemental analyser with a peak
combustion temperature of 900 °C.
From the measured TOC and TN contents the TOC/TN ratio was calculated. This ratio provides information on the sources
and the degradation level of the organic matter (OM) in the sediment, with high values indicating a higher share of terrestrial
source material or well-preserved OM and low values indicating a higher share of aquatic sources or a high level of degradation
of OM (Andersson et al., 2012; Meyers, 1997).
The measurement of the total mercury (Hg) content of the sediment samples was carried out using the direct mercury analyzer
DMA-80 EVO.
The measurement of the $\delta^{13}C$ ratio, as a paleoenvironmental indicator, can also provide information on the sources of OM and
its degree of decomposition. It is mainly determined by photosynthetic processes, but also by other factors like atmospheric
$CO_2$, temperature, and water stress (Andersson et al., 2012). As the first step of the analysis, carbonates were removed from
the samples using hydrochloric acid. Subsequently, the measurement was carried out using a ThermoFisher Scientific Delta-
V-Advantage gas mass spectrometer with a FLASH elemental analyser EA 2000 and a CONFLO IV gas mixing system. The
isotope ratio was determined in relation to the Vienna Pee Dee Belemnite standard [‰ vs VPDB].
**3.2.3 Lipid biomarker analysis**
**Measurement**
Subsamples for lipid biomarker analysis were freeze-dried and homogenised. Lipid biomarkers were then extracted from
approximately 8 g of sample material using accelerated solvent extraction (ASE; ThermoFisher Scientific Dionex ASE 350)
with dichloromethane/methanol (DCM/MeOH 99:1). During extraction, samples were held in a static phase for 20 min at
75.5 °C and 5 MPa. For the subsequent analysis, 5α-androstane as a reference for *n*-alkanes in the aliphatic fraction, and 5*a*-
androstan-17-one as a reference for *n*-alkanols in the neutral NSO-fraction were added. Resolved samples were then
fractionated into an aliphatic, aromatic and NSO fraction using a medium pressure liquid chromatography (MPLC) system
(Radke et al., 1980). Subsequently, the NSO fraction was separated into an acidic and neutral polar fraction by a manual KOH
column separation. In preparation for the measurement the neutral NSO fraction was silylated by adding 50 µl DCM and
50 µl N-Methyl-N-(trimethylsilyl)trifluoroacetamide (MSTFA) and heated at 75 °C for one hour. The measurement of *n*-
alkanes in the aliphatic fraction and *n*-alkanols in the neutral NSO fraction was performed using gas chromatography-mass
spectrometry (GC-MS; Thermo Scientific ISQ 7000 Single Quadrupole Mass Spectrometer with a Thermo Scientific Trace
1310 Gas Chromatograph). The GC-MS system was operated with a transfer line temperature of 320 °C and an ion source
temperature of 300 °C. Ionisation was achieved using an ionisation energy of 70 eV at 50 µA. The full scan mass spectra (m/z
50 to 600 Da, 2.5 scans $s^{-1}$) was analysed using the software XCalibur. The *n*-alkanes and *n*-alkanols were quantified by
comparing their peak areas with those of the internal standards.
**Biomarker indices**
In total, five indices were calculated from the measured lipid biomarker concentrations. Three of these indices, calculated from
the *n*-alkane concentrations, provide information on respective sources of the OC.



The first index was the average chain length (ACL) of $n$-alkanes $C_{23-33}$, calculated following equation 3 where $i$ is the carbon
number and $C$ is the concentration (Poynter and Eglinton, 1990; Strauss et al., 2015).
$$ACL = \frac{\sum i \cdot C_i}{\sum C_i} \tag{3}$$
A change of the ACL can indicate a change of the OC sources and thus a change of input vegetation type to the soil profile
(Schäfer et al., 2016). The long chain odd-numbered $n$-alkanes are mainly produced by terrestrial higher plants like bryophytes
($n$-$C_{23}$ & $n$-$C_{25}$), leaf waxes ($n$-$C_{27}$ to $n$-$C_{29}$), and grasses ($n$-$C_{31}$ to $n$-$C_{33}$) (Haugk et al., 2021; Zech et al., 2010).
The second and third indices are the $P_{aq}$ (ratio of aquatic to terrestrial plant material, equation 4) and the $P_{wax}$ (ratio of terrestrial
plant waxes to total hydrocarbons, equation 5), two ratios that can be used as proxies for the intensity of aquatic influence on
the sediments and to differentiate between aquatic and terrestrial plant input (Thomas et al., 2023; Zheng et al., 2007).
$$P_{aq} = \frac{C_{23} + C_{25}}{C_{23} + C_{25} + C_{29} + C_{31}} \tag{4}$$
$$P_{wax} = \frac{C_{27} + C_{29} + C_{31}}{\sum odd\ C_{23-31}} \tag{5}$$
With the $P_{aq}$, developed by Ficken et al. (2000) it is possible to distinguish between submerged and floating macrophytes, with
values between 0.4 and 1, emergent macrophytes, with values between 0.1 and 0.4, and terrestrial plants, values < 0.1, as a
source for OC in the soil. Since this index and its thresholds were developed in tropical regions, the $P_{wax}$ was additionally used
in this study, as seen in Jongejans et al. (2020). The $P_{wax}$, developed by Zheng et al. (2007), indicates the relative proportion
of waxy hydrocarbons from emergent macrophytes and terrestrial plants to total hydrocarbons (Zheng et al., 2007).
The following two indices are used to provide information on the level of degradation of the OC in the soils. The first index is
the Carbon preference index (CPI) of $n$-alkanes, introduced by Bray and Evans (1961). As a measure of alteration of OC,
values of the CPI decrease with the degradation of OC in the soil (Marzi et al., 1993; Strauss et al., 2015). The calculation in
this study was carried out using the equation introduced by Marzi et al. (1993), with a chain length interval of $C_{23-33}$ (equation

199   6).

$$CPI_{23-33} = \frac{\sum odd\ C_{23-31} + \sum odd\ C_{25-33}}{2 \cdot \sum even\ C_{24-32}} \tag{6}$$
The second index as a measure of level of degradation of OC, introduced by Poynter (1989) is the higher plant alcohol index
(HPA). As a basis of this index, it is assumed that the input ratio of $n$-alkanols and $n$-alkanes into a sedimentary environment
is constant. Therefore, the ratio should depend on the extent of degradation, and since the $n$-alkanols are preferentially degraded
over the $n$-alkanes or degraded to $n$-alkanes due to defunctionalisation, the ratio decreases with ongoing degradation (Poynter
and Eglinton, 1990). The index was calculated using the following equation (7) (Poynter and Eglinton, 1990).
$$HPA = \frac{\sum(n-alkanols\ C_{24}, C_{26}, C_{28})}{\sum(n-alkanols\ C_{24}, C_{26}, C_{28}) + \sum(n-alkanes\ C_{27}, C_{29}, C_{31})} \tag{7}$$
**3.3 Statistical analysis**
The statistical analysis of the data included the analysis of central tendencies of the measured parameters across the different
cores, and the comparison of unfrozen and frozen deposits, as well as saltwater influenced sites, and those not influenced by
saltwater. Central tendencies analysis across the different cores was only applied to the SDLAG, TKL, DLB, and UPL cores,
since the LAG and MAR cores had a too small sample size. After testing and disproving a normal distribution of the data, the
nonparametric Kruskal-Wallis rank sum test was chosen to compare the data of the four different sites. For an additional pair-
wise comparison of cores the Mann-Whitney-Wilcoxon test was used. In addition, it was tested if there are statistically
significant differences between deposits that are influenced by saltwater (MAR, LAG, SDLAG) and deposits that are not
influenced by saltwater (DLB, TKL, UPL) and the frozen (SDLAG, DLB, UPL) and unfrozen (MAR, LAG, TKL) cores, using





the Mann-Whitney-Wilcoxon test. All tests of the central tendency analysis were carried out using R (script in Sect S4.1 &
S4.2 in the supplements).
To test the data for existing correlations between the different measured parameters, a correlation matrix was created in R
(script in Sect. S4.3 in the supplements). The calculation of the correlation was carried out after Pearson. The finished plot of
the correlation matrix only shows correlations with a significance of $p < 0.05$.

## 221 4 Results

### 222 4.1 Sedimentology

The upland permafrost core (UPL) is generally dominated by silt, with a percentage share varying between 63.48 % and
77.5 %. The grain size distribution (GSD) over the whole length of the core is dominated by a peak in the area of fine sands
and silts (figure 2). The sediment samples of the thermokarst lake (TKL) are dominated by silt, with a share ranging between
73.55 % and 80.37 %. The GSD is relatively homogenous over the length of the core, with a peak between silt and clay and a
slight shift towards coarser deposits between 10–16 cm b.s.l. (figure 2). The drained lake basin core (DLB) is dominated by
silt, ranging between 73.31 % and 79.91 %, with sand being represented with only between 2.3 % and 6.7 %. The GSD varies
very little throughout the core, with mean grain sizes between 5.7 µm and 6.74 µm (figure 4) and a peak of the GSD at fine
grain sizes between silt and clay (Figure 2). The GSD of the semi-drained lagoon (SDLAG) has a shift from higher shares of
larger grain sizes peaking in the range of fine sand and silt in the deeper part of the core up to 100 cm b.s.l., to smaller grain
sizes with a peak between silt and clay in the upper part of the core. The deposits of the intact lagoon (LAG) are dominated by
silt and the GSD shows a peak at finer grain sizes between clay and silt (figure 2). The deposits (one sample) of the marine
core (MAR) include a bigger sand portion of 58.5 % and show the coarsest grain sizes among the six studied cores with a mean
grain size of 33.31 µm (figure 4).

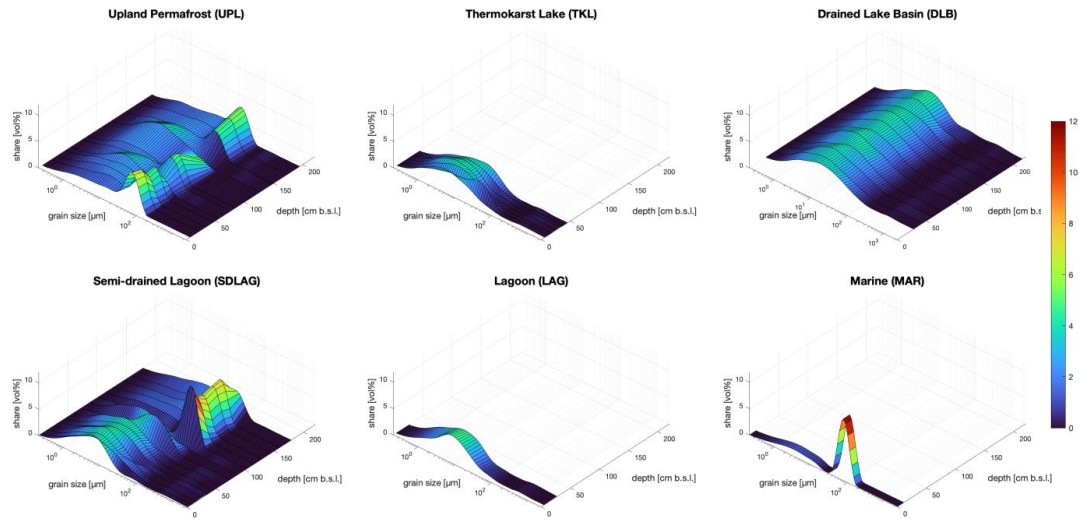


**Figure 2: Three-dimensional grain size distributions over depth [cm] of a land-sea transect**: a) upland Permafrost, b) thermokarst lake,
c) drained lake basin, d) semi-drained lagoon, e) intact lagoon and f) marine profiles. The Colours represent the share [%] of the grain sizes
[µm] with dark blue representing 0 % and red representing 10 %.



**4.2 Biogeochemistry**

The DLB core shows the strongest variations in the TOC content, ranging from 2.94 wt% to 37.62 wt%, with a mean of 7.57 wt% (median 3.26 wt%) (figure 3). The UPL core also shows strong variations in the TOC content, peaking at 20.42 wt% at a depth of 56 cm b.s.l., with a mean of 4.66 wt% (figure 3). In contrast, the TKL sediment core shows a smaller range in the TOC content, between 4.63 wt% and 6.23 wt% (mean 5.37 wt%) (figure 3). It is significantly higher than the TOC content in the upper part of the UPL deposits. The two samples from the LAG plot within the lower end of the range of the TKL deposits, with TOC contents of 4.63 wt% and 4.09 wt% (figure 3). Above 40 cm the TOC content of the SDLAG core varies between those of the UPL and TKL deposits, below it has a consistently lower TOC contents than the other deposits, with a mean of 2.37 wt%, which is significantly lower than in the DLB and TKL samples (figure 3). Additionally, the sample of the MAR deposits has a very low TOC content of 1.3 wt% (figure 3).

The highest TOCvol was determined in the TKL deposits, with a mean of 48.02 kg m$^{-3}$ (figure 4). It is significantly higher than in the SDLAG deposits (mean 32.23 kg m$^{-3}$) and the DLB deposits, with the lowest mean of 25.06 kg m$^{-3}$, both with strong variations in the TOCvol over depth (figure 3 & 4). The strongest variation in the TOCvol is shown by the UPL core, ranging between 6.79 kg m$^{-3}$ and 119.7 kg m$^{-3}$ (figure 3). The mean TOCvol of the UPL deposits of 36.66 kg m$^{-3}$ is relatively high (figure 4). The TOCvol of the marine sample is again relatively low with 20.86 kg m$^{-3}$ (figure 4).

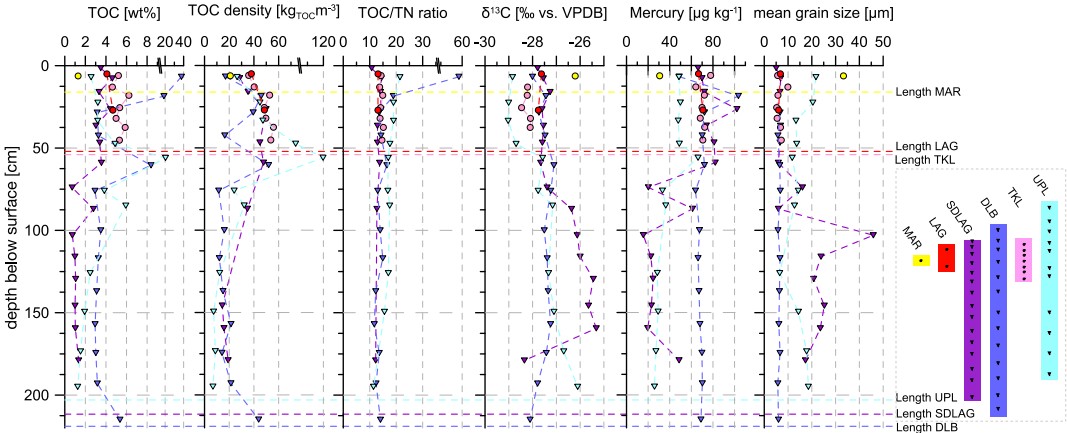

**Figure 3: Summary of the biogeochemical parameters:** total organic carbon (TOC) in weight percent[wt%], TOC density [kg$_{TOC}$cm$^{-3}$], total organic carbon/total nitrogen ratio (TOC/TN ratio), stable carbon isotope ratio ($\delta^{13}$C) per mil relative to Vienna PeeDee Belemnite standard [‰ vs. VPDB], Mercury ]µg kg$^{-1}$] and mean grain size [µm] of the UPL, TKL, DLB, SDLAG, LAG, and MAR profiles, with circles for unfrozen sediments and triangles for frozen sediments. Core abbreviations: UPL: upland permafrost; TKL: thermokarst lake; DLB: drained lake basin; SDLAG: semi-drained lagoon; LAG: lagoon; MAR: marine. Split x axis for TOC, TOC density and TOC/TN ratio.

The TOC/TN ratio is highest in the UPL deposits (mean 17.23), which is significantly higher than in all thermokarst influenced deposits (figure 4). The lowest TOC/TN ratios were measured in both lagoonal sites, with a mean of 13.1 in the SDLAG core (figure 4). The TOC/TN ratios of the SDLAG core are additionally significantly lower than TOC/TN ratios of the TKL deposits, with a mean of 14.39 (figure 4). The DLB core shows the highest ratio of 58.46 in the uppermost sample and a strong decrease in the deeper samples resulting in a mean of 17.5 (median 13.95) (figure 3 & 4). The TN content of the MAR sample below the detection limit resulted in no TOC/TN ratio.

Strongest variations in the $\delta^{13}$C ratio were measured in the UPL ($-26.1$ to $-29$ ‰) and SDLAG ($-25.3$ to $-28.3$ ‰) deposits (figure 3). It is lowest, around $-29$ ‰, in the upper 50 cm of the UPL core and increases in the deeper part of the core (mean $-27.8$ ‰) (figure 3). Both the DLB (mean $-27.5$ ‰) and the SDLAG (mean $-26.9$ ‰) deposits have significantly higher $\delta^{13}$C ratios than the TKL deposits, with the lowest mean $\delta^{13}$C ratio of $-28.2$ ‰ (figure 4).



The mercury (Hg) analysis of the different cores shows that the thermokarst influenced deposits have higher Hg concentrations
compared to the UPL deposits. Significant differences in the Hg content were observed between the DLB and UPL deposits,
as well as between the TKL and UPL deposits, with the UPL samples having significantly lower Hg concentrations (figure 4).
The median Hg content of the TKL samples (70.63 µg kg⁻¹) is nearly twice as high as the median of the UPL samples
(36.34 µg kg⁻¹). Furthermore, the Hg levels of the two samples of the LAG core are in the same range as in the TKL samples
(figure 3). The SDLAG profile shows the largest variations in the Hg content across the samples and has no significant
differences to the other cores (figure 3 & 4).

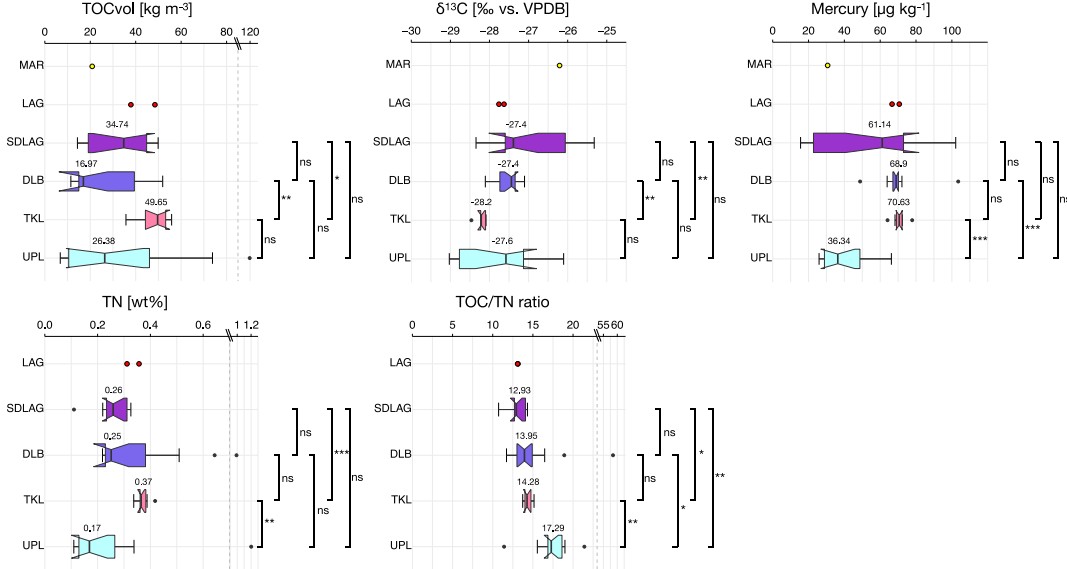


**Figure 4: Boxplots of the biogeochemical parameters**: total organic carbon density (TOCvol) [kg$_{TOC}$cm⁻³], stable carbon isotope ratio
(δ¹³C) per mil relative to Vienna PeeDee Belemnite standard [‰ vs. VPDB], Mercury [µg kg⁻¹], total nitrogen (TN) in weight percent [wt%],
and total organic carbon/total nitrogen ratio (TOC/TN ratio) of the SDLAG, DLB, TKL, and UPL profiles and MAR and LAG as individual
samples. The whiskers display the data range (outliers as black points) and the boxes show the interquartile range (25-75 %). The black
vertical line marks the median and the notches represent the 95 % confidence interval. The bars right of the boxes show the statistical
significance of differences between the profiles (ns = not significant; * = p < 0.05; ** = p < 0.01; *** = p < 0.001). Core abbreviations: UPL:
upland permafrost; TKL: thermokarst lake; DLB: drained lake basin; SDLAG: semi-drained lagoon; LAG: lagoon; MAR: marine. Split x
axis for TOCvol, TN, TOC/TN ratio.
**4.3 Biomarker**
**4.3.1 Organic carbon source indicating indices**
The average chain lengths of *n*-alkanes (ACL) are highest in the three samples of the UPL core, with the highest value of 28.73
in the sample from the middle part (figure 5). Lowest values have been detected for the LAG and the MAR samples, with the
lowest from the MAR core (26.2) at a depth of 6.25 cm b.s.l. (figure 5). All cores with more than one sample show higher
ACL values in deeper part of the core (figure 5).
As shown in figure 5, the highest $P_{aq}$ values were measured in the MAR sample and the uppermost DLB sample, both having
a $P_{aq}$ of 0.66. The MAR sample also has the lowest $P_{wax}$ of 0.52 indicating together with the $P_{aq}$ an aquatic influence on the
OM composition (figure 5). Also, the uppermost DLB sample shows a relatively low $P_{wax}$ of 0.56 (figure 5). Other samples
with high $P_{aq}$ and low $P_{wax}$ are both LAG samples, with a $P_{aq}$ between 0.61 and 0.64 and a $P_{wax}$ between 0.54 and 0.55, and
the uppermost SDLAG sample with a $P_{aq}$ of 0.62 and a $P_{wax}$ of 0.53 (figure 5). The highest $P_{wax}$ values were calculated for
all UPL samples, ranging between 0.76 and 0.74 (figure 5). At the same time, they show the lowest $P_{aq}$ values, varying between



0.31 and 0.39 (figure 5). Another sample with a high $P_{wax}$ of 0.73 and a low $P_{aq}$ of 0.41 is the DLB sample from a mean depth
of 65.25 cm b.s.l. (figure 5). Overall, the data shows two end members, the marine sample with the most aquatic OM source
and the upland permafrost samples with the most terrestrial OM source with the samples from the other location distributed
between the two.

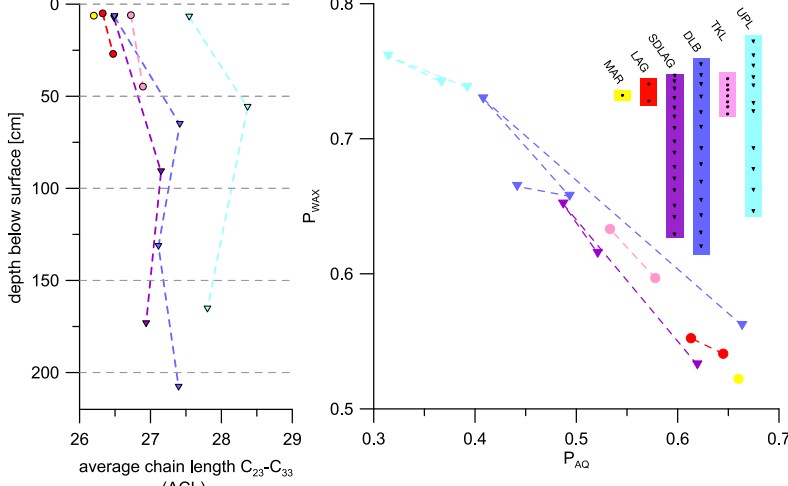


**Figure 5: Plots of the organic carbon sources**, indicated by the n-alkane indices average chain length (ACL) and the proxies $P_{AQ}$, for
aquatic OM, and $P_{WAX}$, for terrestrial OM, with circles representing unfrozen sediments and triangles representing frozen sediments. Core
abbreviations: UPL: upland permafrost; TKL: thermokarst lake; DLB: drained lake basin; SDLAG: semi-drained lagoon; LAG: lagoon;
MAR: marine.

**4.3.2 Organic carbon quality indicating indices**
The carbon preference index of *n*-alkanes (CPI) shows the widest range in the samples of the DLB core, ranging between 7.88
in the deepest sample and the overall highest value of 12.31, calculated for the sample from a depth of 65.25 cm b.s.l. (figure 6).
The lowest CPI values of 5.67 and 6.51 were measured in the LAG samples (figure 6).
The higher plant index (HPA) varies between 0.46 in the deepest SDLAG sample, and 0.81 in the deeper LAG sample
(figure 6). The patterns of the HPA over depths in UPL, DLB and SDLAG samples are similar to the pattern of the CPI in
terms of values increasing or decreasing over depth within each site (figure 6). In contrast, the patterns of the HPA over depth
of TKL and LAG are reversed compared to the CPI, with an increasing value from the deeper sample to the uppermost one
(figure 6).



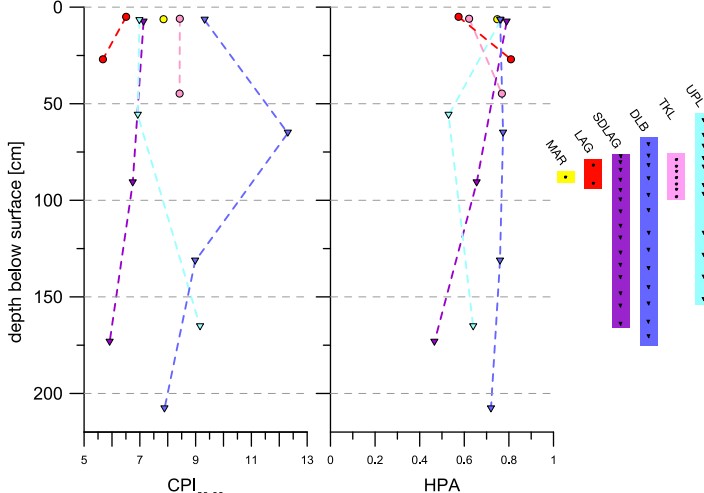

**Figure 6: Plots of the organic carbon quality**: indicated by the lipid-biomarker indices carbon preference index (CPI) and higher plant
index (HPA), with circles for unfrozen sediments, triangles for frozen sediments. Core abbreviations: UPL: upland permafrost; TKL:
thermokarst lake; DLB: drained lake basin; SDLAG: semi-drained lagoon; LAG: lagoon; MAR: marine.
**5 Discussion**
**5.1 Organic carbon**
**5.1.1 Organic carbon characteristics**
The total range of TOC contents, as well as the TOCvol, of all samples is wide (TOC: 0.72–37.62 wt%; TOCvol: 6.79–
119.7 kg m$^{-3}$) (figures 3 & 4), but comparable to other studies that include permafrost and thermokarst features (TOC: 0.2–
43 wt%; TOCvol: 2.8–93.5 kg m$^{-3}$) (Strauss et al., 2015). A reason for this variability is probably the heterogeneity of the
organic source material from the different permafrost and thermokarst landscape features including well-preserved peat,
paleosoils and marine influenced coastal areas. The large range of the TOC content (2.94–37.62 wt%) in the DLB core is likely
caused by such a mixture of permafrost soils and thermokarst lake origin with different material type input and decomposition
processes. Additionally, post-drainage peat accumulation that caused the high TOC contents in the upper soil of the DLB, has
been previously shown in other drained thermokarst lake basin studies as well (Fuchs et al., 2019; Jones et al., 2012; Lenz et
al., 2016). The large, often flat-bottomed drained lake basins provide perfect conditions for the formation of wetlands, through
which most become vegetated in 5–10 years after the drainage event and accumulate peat 10–20 years after (Bockheim et al.,
2004; Jones et al., 2012). Compared to the mean TOCvol of permafrost deposits from the Yedoma region (19 kg m$^{-3}$) and of
thermokarst deposits (33 kg m$^{-3}$) (Strauss et al., 2013), the mean TOCvol of the cores of this study are relatively high (UPL:
37 kg m$^{-3}$; TKL: 48 kg m$^{-3}$; DLB: 25 kg m$^{-3}$; SDLAG: 32 kg m$^{-3}$), revealing a large pool of carbon in all deposits studied (figure
4). The high TOCvol in the TKL deposits, significantly higher than in the SDLAG and DLB deposits, are likely the result of
an interplay of various factors. It might be partially related to the relocation of organic matter (OM) e.g., due to erosion, leading
to OC accumulation in the basin and thaw subsidence progression due to ground ice loss (Lenz et al., 2016). Additionally, it
is likely that there is a higher input of Holocene OC and an increased primary productivity in the lake stimulated by nutrient
release from thawing permafrost (Strauss et al., 2015). The accumulation of OC might be further accelerated by slow
decomposition rates in the cold and anaerobic lake environment (Strauss et al., 2015). The lower TOCvol in the refrozen
thermokarst features (SDLAG & DLB) might partially be influenced by ground ice accumulation after the drainage of the



water bodies. In case of the SDLAG deposits, the lower TOCvol is combined with a low mean TOC content (2.37 wt%), which
might be also influenced by a decrease of the primary productivity with the transition from thermokarst lake to lagoon, since
strong seasonal fluctuations of the salt content, the lowered, fluctuating water level to almost drainage, and the bedfast ice
formation in winter, shortens the period of biological production. Moreover, there might have been decomposition of OM in
the SDLAG deposits all year round when the lagoon had more water or rather was in the state of a thermokarst lake, which
also could have led to a decreased TOC content.
The analysis of the OC and lipid biomarkers in the deposits shows that they contain OM from different sources, likely
additionally influenced by parameters such as salinity, temperature, and water availability. This results in two end members
for the sample set, MAR and UPL, with the other sites aligning between. It nicely depicts the transformation processes of soil
OM over the course of landscape development from dry terrestrial permafrost over thermokarst lakes, saltwater exposure and
finally a marine state (Jenrich et al., 2021). One indicator for the source of OM is the TOC/TN ratio, with lower values
indicating a stronger aquatic influence and higher values indicating a stronger terrestrial influence (Meyers, 1997). The highest
mean TOC/TN ratio was measured in the UPL deposits (17.2), significantly higher than in the three thermokarst landscape
features included in the statistical analysis, indicating the strongest terrestrial influence on the OM composition of the UPL
core (figure 3 & 4). The lowest mean TOC/TN ratios, significantly lower than in the UPL and TKL deposits, were measured
in the LAG and SDLAG samples (13.1), indicating the strongest aquatic influence on those deposits, e.g. from algae and
bacteria. The largest variation of the TOC/TN ratio is shown in the DLB core (11.7–58.5), indicating different sources of OM
during the different stages of the thermokarst lake evolution. Since the TOC/TN ratio can also be influenced by other processes
like the level of degradation of OM, we also analysed the $n$-alkane distribution in the samples and calculated the $P_{aq}$ and $P_{wax}$
as indicators of the source of OM. The results of these parameters also show the two end members (figure 5) with the highest
ACL values and highest $P_{wax}$, thus the strongest terrestrial influence on the OM composition in the UPL deposits and the
strongest aquatic influence on the OM composition in the marine sample, with the lowest ACL and a high $P_{aq}$. It is also shown
in figure 5 that all thermokarst deposits (LAG, SDLAG, DLB & TKL) align between the two end members, thus are stronger
influenced by aquatic OM than the UPL samples. Moreover, figure 5 hints on a change of source of OM in the SDLAG, DLB
and UPL profiles from the upper soil compared to the samples between 50 and 100 cm b.s.l. and between 100 and 200 cm
b.s.l. This might not be influenced by different stages of the thermokarst lake evolution, but rather by changes of hydrological
conditions at the time of deposition, or by the relocation of OM, for example due to cryoturbation or roots, since both the
terrestrial endmember UPL and the thermokarst features show that changes.

### 5.1.2 Organic carbon degradation

On the basis of the TOC/TN and the $\delta^{13}$C ratios, as well as the biomarker indices CPI and HPA the level of degradation of the
OM stored in the soils is discussed (figure 3, 4 & 6).
The decomposition of OM releases carbon as $CO_2$ and $CH_4$ and portions of nitrogen as $N_2O$ from the soils to the atmosphere
(Schuur et al., 2022; Strauss et al., 2024b; Voigt et al., 2020). Deposits containing further degraded OM have lower TOC/TN
ratios than those containing fresh OM due to a larger share of nitrogen in the soils (Andersson et al., 2012; Weintraub and
Schimel, 2005). Thus, in addition to the OM sources the TOC/TN ratios also contain a component dependent on the OM
decomposition level. As seen above, the TOC/TN ratios in the UPL deposits were significantly higher compared to the
thermokarst influenced deposits (SDLAG, TKL, DLB), which was interpreted as a higher terrestrial character of the OM in
the UPL samples. However, it is also likely that parts of the differences derive from the fact that the thermokarst deposits
contain stronger degraded OM, due to longer unfrozen periods. The mean TOC/TN ratio of the UPL (17.23) is in the lower
range of the ratios measured by Routh et al. (2014) in Arctic peat soils (15–25) and lower than the mean TOC/TN measured
by Fuchs et al. (2019) in upland permafrost samples in the Teshekpuk region (21.3). However, they are higher than the mean
TOC/TN ratio measured by Haugk et al. (2021) in Siberia (13.2). The mean TOC/TN ratios in the TKL (14.4) and the DLB





(17.5) profiles are slightly higher than those measured by Fuchs et al. (2019) with a mean TOC/TN ratio in the upper 100 cm
of the soils of 12.6 in TKL deposits and 16.6 of DLB deposits. These rather high values, compared to literature, found in all
profiles indicate a relatively high level of preservation of the accumulated OM, leading to a likely good quality for future
degradation and therefore a vulnerability to decomposition after thaw.
The carbon isotopic signal is also influenced by both factors: OM sources and OM degradation. Terrestrial material usually
shows lighter and marine OM heavier $\delta^{13}$C signals and due to the preferred release of $^{12}$CO$_2$ during degradation, the residual
OM becomes isotopically heavier (Andersson et al., 2012). In the uppermost samples (down to 50 cm) the data resembled the
two-end member model of the OM sources with the UPL samples showing the lightest $\delta^{13}$C values (stronger terrestrial
character) and the MAR sample exhibiting the heaviest signal (marine influenced) (figure 3). The other samples show
intermediate data resembling supply of different OM sources and/or different level of degradation. In the deeper part the picture
is less clear. The UPL samples are isotopically heavier plotting in the range of the DLB data, whose $\delta^{13}$C signal is relatively
constant throughout the whole core. This could indicate a higher level of degradation of OM in the deeper UPL deposits. The
deeper SDLAG samples are, with exception of the deepest sample, isotopically significantly heavier which could indicate a
stronger aquatic/marine influence in the lagoon during time of deposition rather than a stronger degradation of the OM.
Also, the CPI depends on both the source of OM and the level of OM maturation. The original odd-over even carbon number
predominance of the indigenous *n*-alkanes in a sample is determined by the source material and is changing to lower values
during OM maturation. Here, the wide range of different CPIs most likely rather resemble the various mixtures of OM at the
different sites. This is supported by findings of Jongejans et al. (2021), also reporting that the CPI represents rather the source
OM in such relatively young sediments. The HPA shows a very narrow band of values for all samples. In the uppermost sample
of the TKL and LAG, samples show a shift to lower values which could indicate a higher degradation of the OM in the surface
sediments. The UPL and SDLAG samples show lower HPA values in the deeper part of the core which might point to periods
of stronger degradation in the past. However, the material shows low variability in the HPA values overall, plotting in the
upper scale of the parameter and therefore indicating relatively less degraded OM. Thus, with ongoing climate warming and
thawing of the deeper permafrost layers, the preserved OM of good quality could become available to decomposition, leading
to increased emissions of greenhouse gases.
**5.2 Additional Parameters**
Processes that have an influence on OC characteristics in soils can also have effects on other parameters. To identify such
associations, a correlation matrix was computed integrating the measured biogeochemical and sedimentological parameters
(figure 7). TOC content and TOCvol are positively correlated with the Hg content in the samples. In general, sources for Hg,
accumulating in Arctic soils, can be both natural and anthropogenic. Natural sources, contributing to the increase of
atmospheric Hg and subsequent deposition into soils, include boreal forest fires and volcanic activity. Anthropogenic input
has significantly intensified due to industrialization and expanding land use (Jonsson et al., 2017). A reason for the positive
correlation of the TOCvol with Hg is presumably that approximately 70 % of the Hg in the Arctic tundra is derived from
gaseous elemental Hg, which is ubiquitously present in the atmosphere (Obrist et al., 2017). Since the deposition of gaseous
elemental Hg is strongly influenced by the Hg uptake of vegetation, sites with a higher input of OM and therefore higher
TOCvol also accumulate higher levels of Hg bound in the plant matter (Obrist et al., 2017). The Hg content in the deposits is
furthermore negatively correlated with the $\delta^{13}$C ratio. This correlation indicates that there are higher mercury contents in the
deposits with OM from a terrestrial or mixed terrestrial/aquatic source. For example, the marine influenced MAR sample with
the highest $\delta^{13}$C signal shows the lowest HG content and the upper UPL samples with the lower $\delta^{13}$C signal shows higher HG
contents than the lower UPL samples with the higher $\delta^{13}$C signal (figure 3). The same can be observed for the SDLAG samples.
Additionally, the Hg content correlates negatively with the mean grainsize. This is displayed in the mercury contents in the
fine-grained freshwater thermokarst features (mean DLB: 69.87 µg kg$^{-1}$; mean TKL: 70.74 µg kg$^{-1}$) that are significantly





higher than in the UPL deposits (mean UPL: 40.16 µg kg⁻¹) (figure 4). A reason for this could be that the thermokarst processes
might affect the distribution and accumulation of Hg due to the release of Hg from previously freeze-looked Hg-containing
OM in the soil upon decomposition (Schuster et al., 2018). Additionally, thermokarst, erosion and an increased soil water
movement in a thickening active layer, all triggered by permafrost thaw, can increase the transport of Hg from the soils to
Arctic surface waters, resulting in higher Hg concentrations in lacustrine and post-drainage sediments (Rydberg et al., 2010),
which is also indicated by the data of this study. Especially in the SDLAG core the correlation of thermokarst processes with
OC and sediment characteristics and the Hg content is visible. The GSD shows a peak at coarser grain sizes, between fine sand
and silt, similar to the UPL deposits in the deeper half of the core below 100 cm b.s.l. (figure 2). The upper half of the core
shows a peak at finer grain sizes similar to the thermokarst features, indicating lacustrine deposits (figure 2). This shift indicates
that there is less influence of thermokarst processes in the deeper half of the core. Additionally, there are lower Hg contents in
the deeper part (15.57–48.65 µg kg⁻¹) akin to the Hg content in the UPL deposits and accompanied by low TOC contents (0.74–
1.35 wt%) (figure 3). In contrast, the thermokarst influenced upper half of the core show higher Hg concentrations (20.27–
102.17 µg kg⁻¹), similar to the Hg concentrations in the other thermokarst features, accompanied by higher TOC contents
(0.72–4.65 wt%) (figure 3).

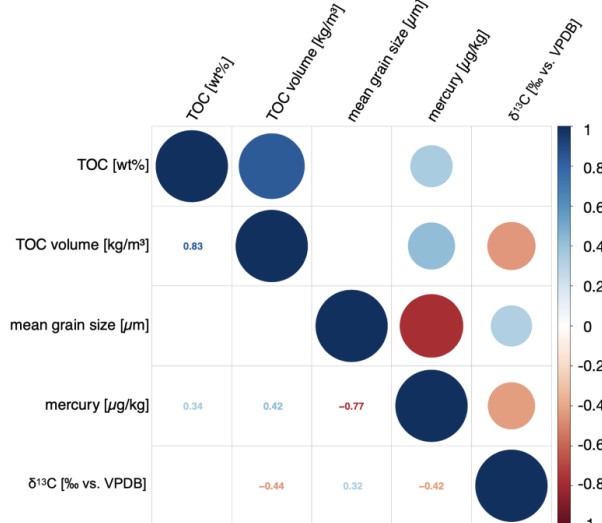


**Figure 7: Correlation matrix of sedimentological and biogeochemical parameters.** Strong positive correlations in dark blue, strong
negative correlations in dark red. TOC: total organic carbon content in weight percent; TOC volume: organic carbon density; $\delta^{13}$C: stable
carbon isotope ratio in per mil relative to Vienna PeeDee Belemnite standard.

**5.4 Influence of salinity and soil condition on the biogeochemical soil characteristics**
The statistical analysis for differences between frozen (UPL, DLB, SDLAG) and unfrozen (TKL, LAG, MAR) as well as
saline (SDLAG, LAG, MAR) and non-saline (UPL, TKL, DLB) deposits shows for most parameters only low variation (figures
8 & 9). However, significant differences were found for ACL, $\delta^{13}$C, TOC/TN ratio and CPI for the saline/non-saline sites and
for TOCvol, TN, ACL and $\delta^{13}$C for the frozen/unfrozen sites.
Both, the comparison of the saline/non saline and the frozen/unfrozen deposits show significant differences for the ACL of *n*-
alkanes. Since the ACL is influenced by the source of OM in the soil, this likely indicates that the input of OM is influenced
by the salinity and whether the soils are frozen or unfrozen. It is significantly lower in the saline (26.48) and unfrozen deposits
(median ACL 26.47) compared to the non-saline (27.4) and frozen deposits (median ACL 27.27) (figure 8 & 9) indicating a
stronger aquatic influence on the OM composition in the saline and/or unfrozen deposits. In case of the comparison of the
saline/non-saline deposits this is accompanied by significantly higher $\delta^{13}$C ratios  and lower TOC/TN in the saline deposits



(median δ13C: –27.47; median TOC/TN: 13.1) compared to the non-saline deposits (median δ13C: –27.58; median TOC/TN:
14.76) (figure 8), supporting the presence of a stronger aquatic OM proportion in the saline deposits. The CPI values are higher
in the non-saline samples, which could resemble different odd over even carbon number predominance distribution of *n*-
alkanes in the aquatic/marine vs. terrestrial organic biomass. All three parameters, the $\delta^{13}C$, the TOC/TN ratio, and the CPI,
might additionally indicate more fresh, undegraded OM in the non-saline deposits, which is likely influenced by a decreased
input of fresh OM in the saline environments due to a decreased primary productivity, an increased microbial activity, since
the salinity in the soil water leads to a depression of its freezing point, thus a longer unfrozen period, and less retention of fresh
OM in the coarse marine sediments (Bischoff et al., 2018; Jongejans, 2022).

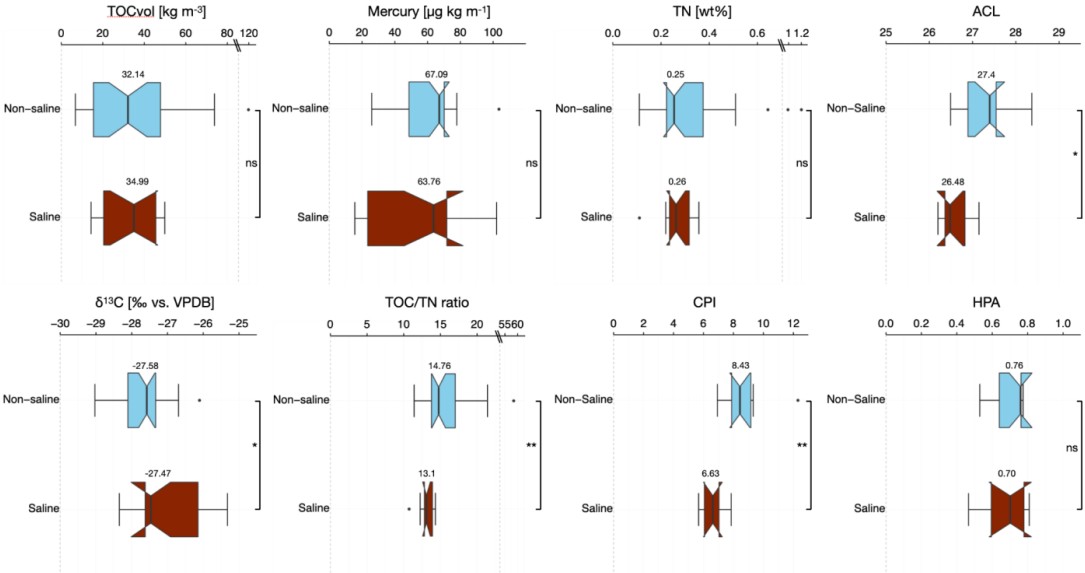


**Figure 8: Boxplots of the biogeochemical parameters divided in saline and non-saline sediments:** total organic carbon density (TOCvol)
[kg$_{TOC}$m$^{-3}$], Mercury [µg kg$^{-1}$], total nitrogen (TN) in weight percent [wt%], average chain length of *n*-alkanes (ACL), stable carbon isotope
ratio ($\delta^{13}C$) per mil relative to Vienna PeeDee Belemnite standard [‰ vs. VPDB], total organic carbon/total nitrogen ratio (TOC/TN ratio),
carbon preference index (CPI), and higher plant alcohol index (HPA) of profiles in non-saline [blue] (including upland permafrost,
thermokarst lake sediments, and drained lake basin sediments) and saline [red] (including semi-drained lagoon sediments, lagoon sediments,
and marine sediments) soil settings. The whiskers display the data range (outliers as black points), and the boxes show the interquartile range
(25–75 %). The black vertical line marks the median and the notches represent the 95 % confidence interval. The bars right of the boxes
show the statistical significance of differences between the groups (ns = not significant; * = $p < 0.05$; ** = $p < 0.01$; *** = $p < 0.001$).
Moreover, the comparison of the frozen and unfrozen deposits shows significant differences in the TOCvol and the TN content.
The frozen deposits have significantly lower TOCvol (median 24 kg m$^{-3}$) and TN (median 0.24 wt%) compared to the unfrozen
deposits (median TOCvol 48.41 kg m$^{-3}$; median TN 0.36 wt%) (figure 9). The higher TN content in the unfrozen deposits is
likely influenced by erosion processes, reactivated soil water movement in thawed permafrost, as well as surface runoff from
nitrogen-rich upland permafrost and the refrozen thermokarst features, leading to the deposition of nitrogen in the aquatic
systems (Strauss et al., 2024b). Furthermore, if bioavailable, the increased TN content in thawed permafrost soils could
potentially enhance the ecosystem productivity, thereby influencing the increased TOCvol in the unfrozen deposits. Also, the
significantly lower $\delta^{13}C$ values in the unfrozen deposits potentially indicates a higher input of fresh OM to the unfrozen
thermokarst environments. Additionally, thaw subsidence progression in the unfrozen deposits and the accumulation of ground
ice in the (re)frozen deposits likely have an influence on the TOCvol (Strauss et al., 2015).



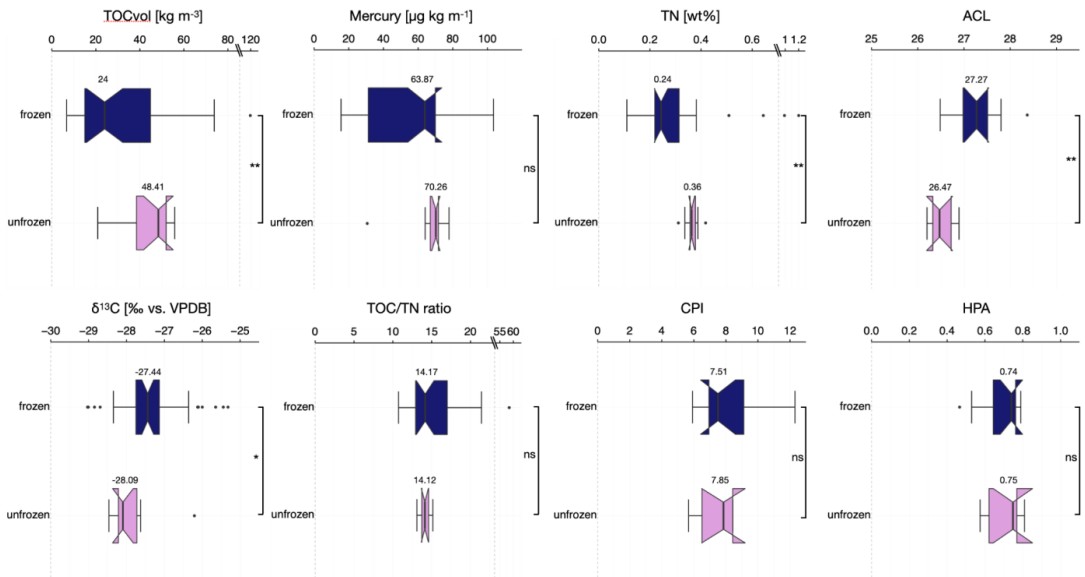

**Figure 9: Boxplots of the biogeochemical parameters divided in frozen and unfrozen sediments:** total organic carbon density (TOCvol) [$kg_{TOC}cm^{-3}$], Mercury [$\mu g\ kg^{-1}$], total nitrogen (TN) in weight percent [wt%], average chain length of *n*-alkanes (ACL), stable carbon isotope ratio ($\delta^{13}C$) per mil relative to Vienna PeeDee Belemnite standard [‰ vs. VPDB], total organic carbon/total nitrogen ratio (TOC/TN ratio), carbon preference index (CPI), and higher plant alcohol index (HPA) of frozen profiles [dark blue] (including upland permafrost, drained lake basin sediments, and semi-drained lagoon sediments) and unfrozen [pink] (including thermokarst lake sediments, lagoon sediments, and marine sediments) soil profiles. The whiskers display the data range (outliers as black points), and the boxes show the interquartile range (25–75 %). The black vertical line marks the median and the notches represent the 95 % confidence interval. The bars right of the boxes show the statistical significance of differences between the groups (ns = not significant; * = $p < 0.05$; ** = $p < 0.01$; *** = $p < 0.001$).

The HPA data are quite similar for the frozen/unfrozen and saline/non-saline sites and plot in the upper range of the parameter scale. This could indicate a comparable level of degradation between all sites and the potential to act as a good substrate for greenhouse gas production when actively metabolized. No significant differences were additionally identified for the Hg content. This might be influenced by the way Hg accumulates in sedimentary deposits. We see evidence that thawing permafrost initiates the reactivation and accumulation of Hg in thermokarst affected deposits. Unlike other measured parameters, these processes are not necessarily reversed upon refreezing of the deposits, but instead tend to pause until repeated thawing of the soils. Consequently, the amount of Hg in the soils is likely to increase with every thermokarst lake and thawing cycle the deposits undergo, without the current soil condition and other properties such as salinity, having a major influence accumulative effect.

## 6 Conclusion

The analysis of the six sediment cores from a thermokarst-affected coastal lowland in North Alaska showed that the OC characteristics in deposits of the different landscape features are diverse. The highest TOC contents were measured in the drained lake basin and thermokarst lake deposits, likely caused by an increased primary productivity and Holocene OC input. This is also reflected by the analysis of the quality of OC, with high CPI values indicating fresh, undegraded OM in both profiles. The deposits of a semi-drained thermokarst lagoon had significantly lower TOC contents than the freshwater-influenced thermokarst deposits. Additionally, there were significant differences in the CPI, $\delta^{13}C$, and TOC/TN ratio between saline and non-saline deposits, indicating a domination of aquatic OM in the saline deposits, and moreover likely indicating a higher level of fresh, undegraded OM in the non-saline deposits. The intrusion of saltwater to the deposits seems to lead to a lower quality of OM in the soils, likely influenced by a lower input of fresh OM due to a decreased primary productivity, and



potentially enhanced by degradational processes. Indicated by the ACL and $P_{aq}$, $P_{wax}$, all thermokarst-influenced deposits
showed a stronger aquatic influence on the OM composition than the upland permafrost deposits. Besides the differences in
the source of OM, the comparison of unfrozen and frozen deposits showed higher TOCvol and TN contents in the unfrozen
deposits. This is also likely influenced by differences in the level of primary productivity, depositional- and degradational
processes. Thus, our findings provide valuable insights into the dynamics of carbon storage and vulnerability to decomposition
in response to environmental changes in a coastal permafrost landscape, since they reflect the complex interplay of
environmental factors, landform characteristics and impacts of climate change on these dynamic Arctic landscapes. The
integration of carbon dioxide and methane emission measurements in further studies could complement the findings and
provide an even more comprehensive picture of carbon fluxes across the geomorphological, hydrological, and ecological
diverse landscapes of Arctic coastal lowlands and the influence of permafrost thaw and saltwater intrusion on the deposits.
*Data availability*. The data used in this manuscript are available online: Biomarkers of sediment cores from a land – shore
transect in the Teshekpuk Lake Region in Arctic Alaska, 2022 (https://doi.org/10.1594/PANGAEA.971595); Hydrochemical
characteristics of sediment cores from a land – shore transect in the Teshekpuk Lake Region in Arctic Alaska, 2022
(https://doi.org/10.1594/PANGAEA.971595); Sedimentological characteristics of sediment cores from a land – shore transect
in the Teshekpuk Lake Region in Arctic Alaska, 2022 (https://doi.org/10.1594/PANGAEA.971244); Biogeochemical
characteristics of sediment cores from a land – shore transect in the Teshekpuk Lake Region in Arctic Alaska, 2022
(https://doi.org/10.1594/PANGAEA.971246)
*Supplement*. The supplement related to this article will be available online.
*Competing interests*. The authors declare that nobody of the author team has any competing interests.
*Author contributions*. J. Strauss, M. Jenrich and F. Giest designed this study. J. Strauss, G. Grosse, and M. Jenrich developed
the overall coring plans for the Perma-X Lagoons field campaign and conducted the fieldwork in 2022. B. M. Jones provided
guidance on site selection, field assistance, and logistical support for the expedition. J. Strauss, M. Jenrich and F. Giest did the
subsampling for all cores. F. Giest carried out the laboratory analyses. K. Mangelsdorf supported the biomarker interpretation.
F. Giest wrote the first draft of the manuscript. All co-authors contributed within their specific expertise to data interpretation
as well as manuscript writing.
*Acknowledgements*. We acknowledge support by the Deutsche Bundesstiftung Umwelt to MJ. BMJ was supported by U.S.
National Science Foundation awards OPP-1806213 and OPP-2336164. We thank Justin Lindemann, Jonas Sernau, Antje
Eulenburg and Mikaela Weiner for their support and assistance in the lab. AWI base funds were used for facilitating the
expedition and laboratory analyses. The Teshekpuk Lake Observatory managed by BMJ was used as a base during the
expedition. We thank Ukpeaġvik Iñupiat Corporation for the logistical support, especially for the fixing of snow machines in
remote areas. We further thank the Iñupiat community for allowing us to do work on their land.

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
