# Peer review of "Organic Carbon, Mercury, and Sediment Characteristics along a land"

_EGUsphere, 2024_

## Author Comment (AC1)

**Review comments on "Organic Carbon, Mercury, and Sediment Characteristics along a land – shore transect in Arctic Alaska" Giest et al.**

Reviewer 1

**Reviewer Comment (RC):** The manuscript titled 'Organic Carbon, Mercury, and Sediment Characteristics along a land – shore transect in Arctic Alaska' by Giest et al. presents results from several sediment cores covering the variable coastal permafrost landscape in northern Alaska. The study comprises of downcore geochemical and biomarker dataset showing differences in quality and sources of organic carbon along the transect and provides insights on carbon decomposition on the variable coastal landscape covering different salinities and thermokarst influence. In its current form, the manuscript is rather heavy to read and could use shortening/sharpening of the text, especially the discussion. Below specific comments and a few suggestions also on how to improve the layout and figures.

**Authors' Reply (AR):** Thank you very much for reviewing our manuscript and providing you very valuable and constructive suggestions for improvement. We increased readability and be more precise in the revised manuscript (to be uploaded in a later stage). Please also find our detailed responses to your specific comments below.

**RC:** Specific comments:

Line 23-26: This reads as a list of results, I suggest adding interpretation or removing from the abstract.

**AR:** Thank you. These lines now read "We found that a semi-drained state of thermokarst lakes features the lowest OC content, and TOC and TN are generally higher in unfrozen deposits, hinting at a more intact state of organic matter." which now connects better to the following lines.

**RC:** Line 45: Use only the abbreviation of IPCC as the abbreviation already explained previously.

**AR:** Thank you, changed accordingly.

**RC:** Line 48: Add a reference for the permafrost temperature increase.

**AR:** We added Biskaborn et al. (2019) to this sentence, where these numbers were taken from.

**RC:** Line 115: Give a temperature for the cooled samples.

**AR:** Thank you, we added temperatures for the frozen samples (-20°C) and for the cooled samples (+4°C) to this sentence.

**RC:** Line 117: Were all the analyses executed at AWI Potsdam? Provide this information e.g., in the first paragraph of this chapter.

**AR:** Thank you. Yes, all analyses were conducted at AWI Potsdam, and since this line is within the first paragraph of chapter 3, we changed the sentence to "...to AWI Potsdam, where all further analyses were conducted."

**RC:** Line 118: This first sentence is not necessary.

**AR:** Thank you, we agreed and removed the respective sentence.

**RC:** Line 122: Add here that these 'other laboratory analyses' include hydrochemical analyses as they are not mentioned anywhere in the actual manuscript.

**AR:** Thank you, we made that sentence more precise now by changing it to "In preparation for further analyses, water for hydrochemical analysis was extracted, and subsequently all samples were freeze-dried, determining their weight before and after this process to calculate water respectively ice content.".

**RC:** Line 132: Refer to the supplementary text S1.1 in this paragraph.

**AR:** Thanks for pointing this missing link out, we added "(Sect S1.1 in the supplements)" to the end of the paragraph.

**RC:** Line 160: This heading is missing a number (same on line 176). I suggest also changing this heading to 'Extraction and Analysis' or something similar.

**AR:** Thank you for pointing this out. Unfortunately, the journal standards only allow for three levels of headings, so we decided to leave these sub-subchapters without heading numbers. However, we followed your suggestion, changing the first subchapter heading to "Extraction, measurement and analysis".

**RC:** Line 165: Define the abbreviation NSO.

**AR:** Thank you very much, this is indeed crucial. We added the explanation to the respective sentence: "...in the neutral NSO (nitrogen, sulphur, and oxygen containing) fraction were added".

**RC:** Line 183: I would advise to remove the 'higher plants' here or rephrase as bryophytes should not be classified as higher plants, and referencing 'leaf waxes' as higher plants does not sound correct.

**AR:** Thanks for pointing out this. We revised the sentence to now read "The long chain odd-numbered $n$-alkanes are mainly produced by terrestrial plants like bryophytes ($n$-C$_{23}$ & $n$-C$_{25}$), grasses ($n$-C$_{31}$ to $n$-C$_{33}$), or originate from terrestrial plant leaf waxes ($n$-C$_{27}$ to $n$-C$_{29}$) (Haugk et al., 2021; Zech et al., 2010).".

**RC:** Line 221. Results. The results are reported in a very detailed manner and could benefit of shortening in places. I advise to report the main/significant results and trends seen in the data and then refer to the supplement/database for more details.

**AR:** Thank you very much, we agree and shortened this section significantly, now reading "The upland permafrost core (UPL) is generally dominated by silt (figure 2), as are the sediment samples of the thermokarst lake (TKL) but with a slightly higher share of silty material. Similar results are present for the drained lake basin core (DLB), with all said cores being homogeneous over the whole length (figures 2, 4).The GSD of the semi-drained lagoon (SDLAG) has a shift from higher shares of coarser grain sizes in the range of fine sand and more silty material found below 100 cm b.s.l., while the upper part reaches towards silty and clayish material. The deposits of the intact lagoon (LAG) are again dominated by silt (figure 2). The deposits, namely one sample, of the marine core (MAR) shows a bigger sand portion of 58.5 %, and represents generally the coarsest grain sizes among the six studied cores (figure 4). For more details, please see the supplementary figures S1 and S2, as well as the published measurement data.".

**RC:** Line 222: Bulk density and water content are mentioned in the methods, but the results are not reported in this paragraph. Add a phrase to this section reporting the main results, or just refer to the database where sedimentological data is made available. The authors could also move the bulk density and water content method section to the supplement as these results do not seem to be the most essential and just refer to them in the main document.

**AR:** Thank you, we agree and decided that bulk density and water content are not essential for this manuscript both in the methods and results. We therefore moved the methods for these parameters to the supplementary material (S1.1.)

**RC:** Lines 223-235: It seems that mostly the cores consist of silt and clays, except the marine one. I would shorten this section to report the main trends, exceptions, and then refer to a table/supplement for detailed results.

**AR:** Thank you, as suggested in your previous comment, we shortened this section drastically and refer to the supplement, published data set, and figures for further details.

**RC:** Line 322. Discussion. I encourage the authors to sharpen the discussion and reduce the result reporting in this section. The text in the conclusions flows well so I would suggest writing the discussion in a more similar style to the conclusion.

**AR:** Thank you, we revised the discussion; please find our changes both in the revised manuscript as well as in the replies to your further comments (ready to be uploaded in a next step).

**RC:** Line 323: I advise to remove this heading and instead split the 5.1.1 in two by separating the discussion on the biomarkers under a separate heading.

**AR:** Thank you! Following your advice, we split this section into "5.1.1 Carbon stocks under various geomorphological influences", focussing on TOC content and stocks, and "5.1.2 Influence of various OM sources" featuring the biomarker analysis.

**RC:** Line 327: The authors write that their data is comparable to other studies, so I advise to add more than one reference study.

**AR:** Thank you for pointing this out. We added more studies to better underline the comparability of our findings.

**RC:** Line 343: Is Strauss et al 2015 the most suited reference for this?

**AR:** Thanks! While Strauss et al. indeed provide some information on this, we also added more references here.

**RC:** Lines 374-375: This is not necessary.

**AR:** Thank you, we agree and removed the respective sentence.

**RC:** Lines 383-387: Here one example where the text could be shortened. I advise to summarise all this information into one sentences or even combining with the last sentence of this paragraph.

**AR:** Thank you for suggesting this, we shortened the paragraph in this place as follows: "The mean TOC/TN ratio of UPL is lower than in comparable sites (Routh et al.(2014, Fuchs et al. 2019), while TKL and DLB show higher values, indicating a relatively high level of preservation of the accumulated OM, leading to a likely high quality for future degradation and therefore a vulnerability to decomposition after thaw".

**RC:** Line 412: More descriptive title would help the reader to navigate the discussion (e.g., title of the chapter 5.4 is informative).

**AR:** Thank you, we changed the title in line 412 to "5.2 Effects of OC characteristics on environmental mercury", but prefer to keep the title for chapter 5.4, which is now chapter 5.3.

**RC:** Line 413: 'Other parameters' here sounds vague, giving an example could be helpful.

**AR:** Thank you, this is indeed very unspecific. We changed this sentence to now read "Processes that have an influence on OC characteristics in soils can also have effects on further parameters, a prominent one being THg." (L XX).

**RC:** Line 448-451: This paragraph is not necessary.

**AR:** Thank you for pointing this out, we removed the respective paragraph.

**RC:** Lines 452-459: Here another example where the authors could consider shortening the text as in its current form it is lengthy when the main message is that high ACL and d13C, and low TOC/TN ratios in saline samples indicate the presence of a stronger aquatic OM proportion.

**AR:** Thank your for summing this up so precisely. We changed this paragraph accordingly, now reading "Looking at the differences in ACL and $\delta^{13}$C, specifically comparing saline, unfrozen deposits with non-saline, frozen deposits, and combining it with the rather low TOC/TN ratios in the saline deposits, which is typical for aquatic OM, it is evident that the saline deposits examined in this study showcase a stronger aquatic influence in their OM composition.".

**RC:** Line 494-502: Add references to this paragraph.

**AR:** Thank you, we added more suitable references.

**RC:** Figures: Fig 2. Increase the font size of the axis titles. I would also consider moving this figure to the supplement.

**AR:** Thank you. We increased the font size as suggested, but prefer to keep this figure in the main manuscript to show the large differences within and between examined deposits.

**RC:** Fig 3 and 4: These are showing in essence the same data, so one of these figures could be moved to the supplement.

**AR:** Thank you. We agree and decided to move figure 4 to the supplementary material, as figure 3 is more intuitive and better shows the variation over the core length.

**RC:** Fig 5 and 6: I suggest combining these two figures to a four-panel figure as they show related data.

**AR:** Thank you. We combined the figures accordingly.

**RC:** Supplement:

Fig. S1. I like how this figure clearly shows the differences between the sampling sites and thus, think that it would be useful to have a version of this as one of the main figures (excluding the detailed soil characteristics info). Perhaps the authors could add this as a panel on Fig. 1?

**AR:** Thank you for this suggestion. Due to the differences in north direction between the maps in figure 1 and the core visualisations in figure S1, we decided to keep them separately, but emphasized to check figure S1 in the caption of figure 1.

---

## Author Comment (AC2)

**Review comments on "Organic Carbon, Mercury, and Sediment Characteristics along a land – shore transect in Arctic Alaska" Giest et al.**
**Reviewer 2**

Accept with moderate revisions

**Authors' Reply (AR):** Thank you very much for your very constructive and valuable review

**Reviewers comments (RC):**

Global comments:

Overall the discussion is too centered on describing the results on not enough on interpreting them, although the conclusion is clear and concise.

**AR:** Thank you. In the revised version (will be uploaded in a second step, including the tracked changes we made) we improved the manuscript basing on your and the other reviewers comments

**RC:** The abstract and discussion have a big part on mercury but there is no mention in the conclusion. Mercury feels like an added measurements without much justification. There is not much background on mercury, especially on the different form of mercury and their risks.

**AR:** Thank you very much for your very valuable review. We added the justification on also including mercury to the revised version, as well as to the conclusion. Thank you for raising this point.

**RC:** In the supplementary methods, L29-45 the authors describe pore-water analysis that are not used or cited in the main manuscript. These results can be combined with the bulk TOC, TN ratios. If these analysis are not used for the interpretation of the results the methods can be deleted.

**AR:** Thank you very much for your very valuable review. In accordance with reviewer 1, we decided to remove water content and bulk density from the main manuscript. However, since these data might be interesting to some readers, we decided to keep them, along with the porewater analysis, in the supplementary material, which then requires including the respective methodology.

**RC:** Detailed comments:

L16: "their thawing could lead to the release…" I think that multiple studies have shown that thawing permafrost does release GHG, you should reformulate the sentence, maybe removing "could".

**AR:** Thank you, changed accordingly.

**RC:** L16-17: I guess you did not do any modeling of future impacts of permafrost thaw in this paper, this sentence "warming. To enhance predictions of potential future impacts of permafrost thaw" is overselling the study. Rather you studied the impact of thermokarst processes, which in turn could be useful for predicting future effect of permafrost thaw.

**AR:** Thank you, we fully agree and changed the sentence to now read "To enhance predictions of potential future impacts of permafrost thaw, a detailed assessment of soil characteristics' changes in response to thermokarst processes in permafrost landscapes is needed, which we investigated in this study in an Arctic coastal lowland." (L XX).

**RC:** L22-23: As it is the abstract I would shorten to "lipid biomarkers (n-alkanes, n-alkanols and their ratios).

**AR:** Thank you, changed as suggested.

**RC:** L48: A reference is missing

**AR:** Thanks for pointing this out, we added a suitable reference here.

**RC:** L56: rather than "quality" you analyse the source of OM. If you refer to quality then you need to explain what you mean: lability?

**AR:** Thank you for specifying this. We added "...quality, hence the degree of decomposition, of organic matter (OM) in the different soils as well as differences in OM sources, lipid biomarkers can be used" (L XX) to the respective sentence. However, we disagree that only the OM source is investigated. This holds true for some of the ratios used (ACL, Pwax, Paq), while, if the previously mentioned ratios reveal the presence of similar material, CPI and HPA are used to determine the degree of degradation.

**RC:** L73-74: In this paragraph you talk about mercury in general without identifying if it is total mercury, methylmercury or elemental mercury that is measured. This is important when

stating the danger to human health and ecosystems as the different mercury form will be more or less dangerous. It should be also mentioned that there is usually a correlation between TOC and total mercury (e.g. Chakraborty et al., 2015).

AR: Thank you, this is absolutely right. Basing on your comment we adjusted the paragraph

RC: L76: OM rather than "OC"

AR: We agree, changed accordingly, thank you for this suggestion.

RC: L109: maybe marine coastal instead of just "marine"?

AR: Great suggestion, thanks, changed accordingly.

RC: L153: I guess you measured d13C on bulk organic matter? If so it would be good to add "The measurement of the δ13C signature of organic matter"

AR: Thank you, we changed this to "The measurement of the $\delta^{13}$C signature of bulk organic matter, …" (L XX).

RC: L155: degradation is also a process affecting d13C, as you mention L391,

AR: Thank you. This sentence already says "...can provide information on the sources of OM and its degree of decomposition", which in our opinion already shows the link to degradation.

RC: L153-158: Which reference standard were used to calculate the ratio? What was the instrumental error/accuracy?

AR: Thank you for emphasizing such details. Vienna Pee Dee Belemnite (VPDB) was used as a reference standard, as is already written in the last line of this paragraph. We added the instrumental accuracy of ± 0.15 ‰ here as well.

RC: L190-192: Paq has been used a lot and has many bias, especially in region where floating and submerged vegetation are not fully characterized, such as in the Arctic. In addition, it is rarely used in soils but rather in lake sediment cores. Since the authors are studying different types of environments: soil, coastal, lake, I feel like this ratio does not add much information

as you would expect higher aquatic influence in the marine and lacustrine environment. Similarly for Pwax. In comparison, the degradation indices are much more useful for the study.

**AR:** Thank you very much for your comment. We agree that the indices have their weaknesses. For a multiproxy approach like ours, we think this adds a puzzle piece more. In this case it is useful to see that the lagoon and lake show an aquatic signal, and the upland a terrestrial signal.

**RC:** Fig. 3: d13C needs error bars

**AR:** Thank you for this comment, but we would like to keep it as is for two reasons: First, in such a graph, adding very small horizontal lines to each data point would deem the graph unreadable. Second, since such error bars would only reflect measurement accuracy, which would also be true for most other parameters, they would not increase the value of this graph when the accuracy is reported before. For detailed and accurate information on individual data points, we believe interested readers will anyway consult the measurement result table, published either online or in the supplementary material.

**RC:** L259: small typo "Mercury [ug …]"

**AR:** Thank you, fixed.

**RC:** L382-383: This sentence is very hard to understand, could you rephrase it?

**AR:** Thank you, we tried to make this more clear and linking to the preceding sentence, now reading "However, these differences are likely also influenced by OM degradation during unfrozen periods of the thermokarst deposits." (L XX).

**RC:** L412. The title of this section is quite hard to guess could you make it more detailed (e.g. Mercury content)? In this paragraph the biomarkers are not compared to mercury content, why not? Right now this paragraph does not seem linked to the rest of the study

**AR:** Thank you, as also requested by reviewer 1, we changed this section title to "Effects of OC characteristics on environmental mercury".

**RC:** L415: can you add the correlation factor between TOC and mercury in the text to make it directly visible to the reader. O.34 and 0.42 are quite weak correlation. I would rather point

and explain the correlation with grain size (-0.77). Further down in the text, the correlation with d13C is also rather weak (-0.42) and should be mentioned directly in the text.

AR: Than you, changed accordingly

RC: L427: not just "additionally" but rather "mainly". See above

AR: Thank you; we implemented the suggested change as a result of our changes due to your previous comment.

RC: Figure 7: you can remove the 1:1 correlation point to make the figure more readable

AR: Thank you, we followed your suggestion and removed the blue points in the 1:1 correlation row. Now included as figure 5 in the updated manuscript (basing on the reviewer 1 comment placing figure 4 in the supplement and combining former figure 5 and 6)

RC: L455-456: The ACL difference is mainly indicative of additional primary production on site no? Since the concentration of the alkane is known the authors can mention which alkane increase in the saline and unfrozen deposit compared to the others.

AR: In accordance with comments raised by reviewer 1, we removed the detailed result reporting from this section. Detailed results can be found in the published dataset. However, differences in ACL can also relate to different (plant) species in the different locations. Hence, we prefer to not go into speculative detail here on which of the aforementioned processes affects the ACL in this case. However, we agree that in a manuscript with a stronger focus on biomarkers instead of landscape development, such a comparison would indeed be interesting.

RC: L458: "13" should be superscript

AR: Thanks for pointing this out. However, the respective sentence is not part of the revised manuscript anymore.